# Periprosthetic Stress Shielding of the Humerus after Reconstruction with Modular Shoulder Megaprostheses in Patients with Sarcoma

**DOI:** 10.3390/jcm10153424

**Published:** 2021-07-31

**Authors:** Sebastian Klingebiel, Kristian Nikolaus Schneider, Georg Gosheger, Thomas Ackmann, Maximilian Timme, Carolin Rickert, Niklas Deventer, Christoph Theil

**Affiliations:** 1Department of Orthopedics and Tumor Orthopedics, University Hospital Muenster, 48149 Münster, Germany; Kristian.Schneider@ukmuenster.de (K.N.S.); Georg.Gosheger@ukmuenster.de (G.G.); Thomas.Ackmann@ukmuenster.de (T.A.); Carolin.Rickert@ukmuenster.de (C.R.); Niklas.Deventer@ukmuenster.de (N.D.); Christoph.Theil@ukmuenster.de (C.T.); 2Institute for Legal Medicine, University Hospital Muenster, 48149 Münster, Germany; Maximilian.Timme@ukmuenster.de

**Keywords:** sarcoma, shoulder, shoulder arthroplasty, megaprosthesis, aseptic loosening, radiotherapy, chemotherapy, implant failure, radiolucency, bone resorption, osteolysis

## Abstract

(1) Background: Modular megaprosthetic reconstruction using a proximal humerus replacement has emerged as a commonly chosen approach after bone tumor resection. However, the long-term risk for revision surgery is relatively high. One factor that might be associated with mechanical failures is periprosthetic osteolysis around the stem, also known as stress shielding. The frequency, potential risk factors, and the effect on implant survival are unknown. (2) Methods: A retrospective single-center study of 65 patients with sarcoma who underwent resection of the proximal humerus and subsequent reconstruction with a modular endoprosthesis. Stress shielding was defined as the development of bone resorption around the prosthesis stem beginning at the bone/prosthesis interface. The extent of stress shielding was measured with a new method quantifying bone resorption in relation to the intramedullary stem length. All patients had a minimum follow-up of 12 months with conventional radiographs available and the median follow-up amounted to 36 months. (3) Results: Stress shielding was observed in 92% of patients (60/65). The median longitudinal extent of stress shielding amounted to 14% at last follow-up. Fifteen percent (10/65) showed bone resorption of greater than 50%. The median time to the first radiographic signs of stress shielding was 6 months (IQR 3–9). Patients who underwent chemotherapy (43/65) showed a greater extent of stress shielding compared to those without chemotherapy. Three percent (2/65) of patients were revised for aseptic loosening, and one patient had a periprosthetic fracture (1/65, 1.5%). All these cases had >20% extent of stress shielding (23–57%). (4) Conclusions: Stress shielding of the proximal humerus after shoulder reconstruction with modular megaprosthesis is common. It occurs within the first year of follow-up and might be self-limiting in many patients; however, about one third of patients shows progression beyond the first year. Still, mechanical complications were rare, but stress shielding might be clinically relevant in individual cases. The extent of stress shielding was increased in patients who underwent perioperative chemotherapy. Stress shielding can be quantified with an easy method using the stem length as a reference.

## 1. Introduction

The proximal humerus is a common location for bone malignancies and around 7–10% of bone sarcomas are located there [1,2]. Depending on tumor histology, surgical treatment usually consists of a wide resection [3]. Nowadays, a high percentage of patients with sarcoma can undergo limb-sparing tumor resection and due to the advancements in adjuvant treatment with multiagent chemotherapy regimes and local radiation, around 70% of patients survive in the long term [4]. However, surgery will result in a segmental, juxta-articular bone defect. Orthopedic surgeons are challenged with choosing a durable and stable reconstruction that reinstitutes as much shoulder and arm function as possible. Modular megaprosthetic reconstruction using a proximal humerus replacement (PHR) has emerged as a commonly chosen approach due to its wide availability, early stability, and modularity that allows us to address different anatomies and a variety of defect sizes with relative ease [5,6]. However, in the long term, these implants demonstrated a fairly high failure rate [7,8,9,10]. At the proximal humerus, soft tissue failure is the most common failure mode while mechanical complications such as aseptic loosening and failure of the supporting bone are considered to be rather uncommon [11].

However, one factor that might be associated with long-term mechanical failures is bone rarefaction or osteolysis around prosthetic stems, also known as stress shielding, which has been described in non-megaprosthetic shoulder arthroplasty [12]. However, it has not been investigated for PHR following sarcoma resection [13,14,15,16,17]. Considering that most patients will undergo routine follow-up radiographs, as part of their oncological aftercare, new onset, or progressive osteolysis can cause severe anxiety for the affected patients and may be a cause of concern for radiologists and non-specialized surgeons who are confronted with stress shielding. However, only one study has most recently drawn attention to this phenomenon. Braig et al. reported stress shielding around cemented stems in 23% of cases including 39 patients of whom 67% had PHR due to bone metastases [18]. To our knowledge, there are no studies on stress shielding around uncemented PHR and no studies that focus on primary sarcoma patients despite the fact that for long-term surviving patients, implant survival and associated revision surgeries become more important [19,20].

This study investigates the prevalence of stress shielding around uncemented PHR in the form of complete osteolysis, potential risk factors for its occurrence, and clinical consequences such as aseptic loosening and associated revision surgeries. Furthermore, we provide a tool to quantify periprosthetic stress shielding.

## 2. Materials and Methods

This is a retrospective single-center study of 113 patients who underwent resection of primary sarcoma of the proximal humerus and were treated with reconstruction of the humeral bone defect and glenohumeral joint using modular endoprosthesis between 2000 and 2019. After approval of the local ethics committee was obtained (ethical approval code: 2020-898-f-S), a database analysis of our institution’s electronic patient records was performed. This study was conducted according to the principles of the World Medical Association (Declaration of Helsinki).

We identified 65 patients (40 males and 25 females) with primary sarcoma resection of the proximal humerus and subsequent implantation of PHR using a single-design implant system (MUTARS^TM^—modular universal tumor and revision system, Implantcast GmbH, Buxtehude, Germany) who met the inclusion criteria. The inclusion and exclusion criteria are shown in Figure 1 (strengthening the reporting of observational studies in epidemiology (STROBE) diagram). The median age at the time of tumor resection was 18 years (IQR 14–42).

In 40 (62%) cases, intra-articular resection was performed, and in 25 (38%) cases, extraarticular resection was performed due to intra-articular tumor contamination. During the study period, no other modular systems were used and megaprosthetic reconstruction was the approach of choice for segmental defects adjacent to the shoulder joint. During the earlier years of the study period, cemented stem fixation was preferred; however, from 2006 onwards, a hydroxalapatite-coated stem was preferred for all patients with primary bone sarcomas who presented with good intraoperative bone quality as determined by the surgeon.

For preparation of the humerus after wide tumor resection, the medullary cavity was first deepened using the medullary cavity reamer. The medullary canal was then expanded with a humeral drill 1 mm below the dimension of the preoperatively determined humeral stem. Then, the cavity was prepared successively with a rasp until stable pressfit anchorage was achieved. The stem size is equivalent to the last rasp size. The stem used had a length of 75 mm and a mean diameter of 12 ± 2 mm. The mean reconstruction length was 160 ± 40 mm resulting in a ratio of reconstruction length to stem length of 2.2. For patients with intra-articular resections during which a majority of the deltoid muscle and the axillary nerve could be spared, a reverse prosthesis design using a polyethylene glenosphere was used in in 15 (23%) cases; otherwise, anatomic reconstruction with implantation of a hemiarthroplasty 50 (77%) was performed. All patients had soft tissue refixation using an attachment tube (Trevira^TM^, Secaucus, NJ, USA). The treated extremities were postoperatively immobilized in a brace for 6 weeks, while physiotherapeutic treatment for the adjacent joints was allowed. All patients were evaluated by the local tumor board and depending on tumor histology, adjuvant treatment was performed according to the study protocols of the time. Forty-three patients (66%) received chemotherapy, and 16 patients (25%) had local radiation treatment. (Table 1, patient demographics and study specifics).

Postoperatively, patients underwent a standardized orthopedic and oncological follow-up protocol. The first postoperative radiological examination was performed within the first days after surgery. An orthopedic and oncologic follow-up with imaging of the lung and radiographs in two planes of the affected proximal humerus were performed in three-month intervals during the first two years following completion of oncological treatment. Until the fifth year after surgery or completion of chemotherapy, the intervals were extended to half a year. After five years, annual controls were scheduled. The median follow-up period amounted to 36 months (IQR 18–80) for all patients. As of January 2021, there were 41 surviving patients (63%, median FU of 48 months (IQR 24–83) and 24 patients (37%, median FU of 20 months (IQR 18–74)) who died of disease. The American Shoulder and Elbow Surgeons (ASES) and Musculoskeletal Tumor Society (MSTS) scores were assessed at the latest follow-up visit to determine functional outcome.

Stress shielding was defined as new-onset bone resorption on the level of prosthesis stem beginning at the proximal bone/prosthesis interface. The radiographs taken during follow-up were compared to the initial radiographic findings. In order to determine the extent of the stress shielding, the radiograph with the greatest extent of stress shielding was analyzed, and the distance from the proximal end of the stem to the point of complete bone resorption was measured. This was divided by stem length multiplied with 100 to calculate the percentage of bone resorption (Figure 2, method of measurement of stress shielding). Since stress shielding is defined as a relative parameter, it is independent of scale or calibrated measurements.

In addition, reasons for implant failure leading to revision surgery were analyzed. Failures were defined as proposed by Henderson et al. as soft tissue failure (type I), aseptic loosening (type II), structural failure (type III), infection (type IV), and tumor progression (type V) [11]. Considering that stress shielding might weaken the strength of implant-bone fixation, mechanical failures (types II and III) were of particular interest in this study.

Statistical analysis was carried out with SPSS (Version 27, IBM, Endicott, NY, USA). Depending on data distribution, (non-)parametric analysis was performed with the Mann–Whitney U-test or Student’s t-test. The chi-squared test was applied for categorical variables with a significance level set at *p* < 0.05.

## 3. Results

Stress shielding was observed in 92% of all patients (60/65) at the latest follow-up (median 36 months (IQR 20–82) (Table 2, overview extent stress shielding in different patient groups). At the latest follow-up, the median longitudinal extent of stress shielding (Figure 2) amounted to 14% (IQR 8–25) relative to the stem length in all patients. Fifteen percent of all patients (10/65) showed an extent of bone resorption greater than 50%. Revision-free survivors showed a median stress shielding of 13% (IQR 8–25). The median time to the first radiographic signs of stress shielding was 6 months (IQR 3–9). At this time, the median extent of stress shielding amounted to 5% (IQR 4–9). Seventy-four percent (48/65) of all patients presented signs of stress shielding within the first 12 months of radiological follow-up. In these patients, the median extent was 10% (IQR 6–14) at the one-year follow-up. A progression of stress shielding of at least 10% (mean 28%, range 10–76) was observed in 34% of patients (22/65) after the first year. In the remaining patients, the extent of stress shielding progressed by 3% until the last follow-up.

When investigating risk factors, it was found that patients who had undergone chemotherapy (43/65) had a greater extent of stress shielding compared to patients without chemotherapy (16% (IQR 10–40) vs. 10% (IQR 6–14), *p* = 0.012). In contrast, patients who underwent radiation treatment (16/65) presented a lower extent of stress shielding compared to patients without radiation (7% (IQR 5–21) vs. 14% (IQR 9–28), *p* = 0.08). Furthermore, male patients did not have a significantly greater extent of stress shielding compared to females (median 14% (IQR 10–45) vs. median 11% (5–21), *p* = 0.06). On the other hand, there was no correlation between extent of stress shielding and the duration of follow-up (*p* = 0.17). With the numbers available, there was no difference regarding the extent of stress shielding between patients with extra-articular resections compared to intra-articular resections (17% (8–24 vs. 12% (7–26), *p* = 0.43) and with respect to the use of reverse total shoulder arthroplasty compared to anatomical hemiprostheses (14% (IQR 10–29) vs. 12% (6–21), *p* = 0.47). Additionally, the ratio of stem length to the length of extramedullary reconstruction was not higher in patients with an extent of stress shielding > 50% (1.93 (IQR 1.61–2.43) vs. 2.26 (IQR 1.87–2.53), *p* = 0.46). With the numbers available, postoperative MSTS and ASES scores did not correlate with extent of stress shielding (*p* > 0.05).

During the follow-up period, 11 patients (17%) underwent revision surgery for prosthetic failure. At the last follow-up, 3% (2/65) of patients had undergone revision surgery for aseptic loosening as type II failure (at 55 and 75 months, respectively). Stress shielding was noticeable in both cases amounting to 45% and 23%, respectively. Both patients received a stem exchanged for a cemented stem and have had an uncomplicated follow-up since then (54 and 24 months after surgery). One traumatic periprosthetic fracture (type III failure, 1/65; 1.5%) was sustained by an 11-year-old boy following a fall onto his elbow. The stress shielding at this time (22 months after prosthesis implantation) was 57%. The patient was successfully treated conservatively with a cast. Whether this was a sufficient trauma or a minor trauma can no longer be determined on the basis of the protocols. Unfortunately, the patient died due to progressive disease.

## 4. Discussion

This study analyzes the radiographic prevalence of periprosthetic stress shielding and the associated longitudinal extent in relation to the prosthetic stem after joint reconstruction with cementless PHR in patients following primary sarcoma resection. The most important finding of this study was that the vast majority of patients showed radiographic signs of stress shielding with an extent of around 1/7th of the intramedullary stem length. Higher degree stress shielding (>50% extent) was found in merely 15% of patients. Revision surgeries for mechanical failure or loosening were rare. With the numbers available, it was not possible to present a clear pattern regarding the occurrence or progression of stress shielding, but two-thirds of patients showed no to little further progress after the first year postoperative and the following regular controls. It appears that patients who underwent chemotherapy as part of their multimodal treatment were at a greater risk for stress shielding compared to patients without chemotherapy.

As stress shielding has only been investigated sporadically, to our knowledge, there is no universally accepted method to quantify and measure periprosthetic stress shielding, particularly around modular endoprostheses. Previous studies that investigated stress shielding in non-megaprosthetic upper or lower extremity prosthetic reconstructions have focused on various radiographic changes such as radiolucency, condensation lines, spot welds, or osteopenia to generate a score that is intended to describe the sum of radiographic changes [21]. However, as the frequency and clinical relevance of stress shielding is still evolving, a simple method to quantify periprosthetic bone resorption was developed for this research and used measuring the extent of stress shielding in relation to the intramedullary stem length, therefore eliminating the need for calibrated radiographs. However, considering the rarity of modular PHR, other scoring systems might be more appropriate in conventional shoulder arthroplasties [17,22,23]. Future studies should evaluate and compare which system is most reliable and can be used reproducibly.

The reconstruction of bone defects after resection of malignant bone tumors using megaprosthetic PHR has become a frequently used approach with favorable results in several studies despite potential long-term complications [6,20,24,25,26]. However, considering that this procedure is still relatively rare, there are few studies that investigate individual failure modes, particularly aseptic loosening or failure of the bone–implant interface [11,27,28]. To our knowledge, only one study by Braig et al. analyzed stress shielding after implantation of modular endoprosthesis of the humerus [18]. This study included 39 patients (metastatic disease *n* = 26; primary sarcoma *n* = 7; hematological disease *n* = 5) after proximal humerus resection and reconstruction with cemented modular endoprosthesis focusing on stress shielding, associated risk factors, and perioperative complications. Stress shielding was defined as any periprosthetic bone resorption compared to immediate postoperative radiographs. At a mean follow-up period of 29 months, 23% (*n* = 9) of patients showed signs of stress shielding. Braig et al. conclude that stress shielding might be an underreported problem in megaprosthetic reconstructions of the proximal humerus. In their series, stress shielding was associated with use of short intramedullary implants and larger megaprosthetic reconstruction length. Revision surgery or implant failure due to stress shielding was not reported, however. Compared to our findings, the prevalence of stress shielding was considerably lower in their study (23 vs. 92%). However, due to the shorter follow-up, different and more heterogeneous patient cohorts with significantly increased patient age of 61 ± 16 years and correspondingly different bone quality, use of cemented stems, and another implant design, the results are not comparable. Furthermore, with longer follow-up it would be interesting to see the further course of stress shielding in patients with cemented stems; although, in a mainly metastatic patient cohort, many patients might die of their disease after a relatively short time. Furthermore, the effect of chemotherapy was not investigated in this context. Future studies are needed to compare cemented and uncemented stems with a particular focus on sarcoma patients facing the expected improving long-term survival.

While there are few data available on stress shielding around megaprosthetic stems, this issue has been investigated in multiple studies on conventional shoulder arthroplasty [15,29,30]. Denard et al. compared stress shielding between 58 standard stems and 56 short stems following (anatomical) total shoulder arthroplasty (TSA) [12]. Cortical thinning was present in 74% of the standard stems and in 50% of the short stems. In both groups, calcar osteolysis was described in more than 20%. The pattern of osteolysis was associated with stem design in their study, but still there were no cases of loosening, stem migration, or structural failure after a mean follow-up of 24 months [12]. Additionally, Denard et al. analyzed 93 uncemented press-fit stems and 26 cemented stems in a single design reverse shoulder arthroplasty (RSA) regarding stress shielding and functional outcome [31]. Calcar osteolysis was present in 43% and 58% of cases, respectively. However, there were no differences in function and the risk of loosening. In their studies, Denard et al. calculated a score from different radiographic findings (e.g., spot welds, cortical thinning, osteopenia) at different regions of the stem [32]. Due to the fundamental differences in stem design and the loss of important anatomical landmarks due to tumor resection, this system cannot be applied to our patient group receiving modular megaprostheses.

Nevertheless, our data emphasize that the incidence of stress shielding is a common phenomenon following cementless PHR with an incidence of up to 92%. The median extent of bone resorption in relation to the intramedullary implant length amounted to 14%. While it is questionable that a relatively small amount of stress shielding might be irrelevant from a biomechanical point of view, there are no studies on this matter, and surgeons must consider that there might be relevant levering forces that act on the remaining stem covered by bone. Nonetheless, aseptic loosening or periprosthetic fractures are still very rare [11]. In our study, two of 65 patients presented aseptic loosening after 55 and 75 months, respectively, with an extent of stress shielding of 23% and 45% well above the average of 14%. Likewise, the patient with a periothetic fracture showed an extent of stress shielding of 57%. Therefore, while there are inadequate numbers for statistical analysis, there might be a theoretically increased risk of mechanical failure in patients with extensive stress shielding. Patients should be educated about this potential risk, and future studies should investigate the biomechanically tolerable extent of minimal stem fixation.

In the present study, stress shielding occurred early on within the first three to nine months postoperatively. Seventy-five percent of patients showed progression to 10% within the first year, so that stress shielding appears to be an early onset postoperative phenomenon with initial relatively swift dynamics. The further progress appears to be slower while also non-linear. After the first year, two thirds of patients presented with an average progression of 3% during the following 36 months. It is possible that many patients may reach a steady state after the first year; although, the numbers are too small to reliably comment on this issue.

Considering potential risk factors, patients who underwent chemotherapy had a greater extent of stress shielding compared to patients without chemotherapy. A possible reason might be an increased bone turnover and decreased remodeling potential of the osteoblasts due to chemotherapy [33,34,35]. Therefore, chemotherapy might be a risk factor for stress shielding.

The results of this research have to be interpreted in light of certain limitations. The retrospective design of the study inherently limits the interpretation of the data, we were able to collect and of the performed analysis. It should be noted that individual cases are very heterogeneous with respect to bone resection length due to the tumor involvement and related loss of soft tissue, as well as perioperative oncologic treatment. Due to the overall rather small patient group, we are not able to analyze and compare larger subgroups due to the complexity of the individual cases and many individual variables. On the other hand, we present the largest collective after primary sarcoma resection of the proximal humerus in young patients and a minimum follow-up of one year. We present for the first time a well-reproducible reference method for quantifying stress shielding in a large collective. This simplifies the communication in the clinical routine. Furthermore, all patients were treated using a single-design implant system.

## 5. Conclusions

Stress shielding of the humerus is common around the proximal prosthesis stem following reconstruction with megaprostheses and occurs early postoperatively. It might be self-limiting in many patients; however, about one third of patients shows progression beyond the first year. Still, mechanical complications appear to be very rare, but might be clinically relevant in individual cases. The extent of stress shielding might be increased in patients who underwent perioperative chemotherapy. While this study demonstrates mid-term results, the long-term effects of stress shielding on implant stability are unknown. There is a need for future long-term studies on stress shielding in patients with primary sarcoma or metastatic bone tumors evaluating stem fixation and adjuvant treatments.

## Figures and Tables

**Figure 1 jcm-10-03424-f001:**
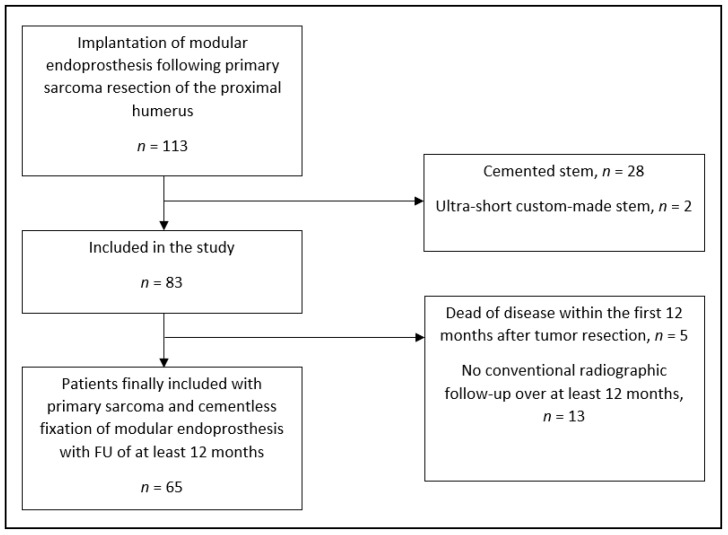
STROBE diagram of the study population included in this study.

**Figure 2 jcm-10-03424-f002:**
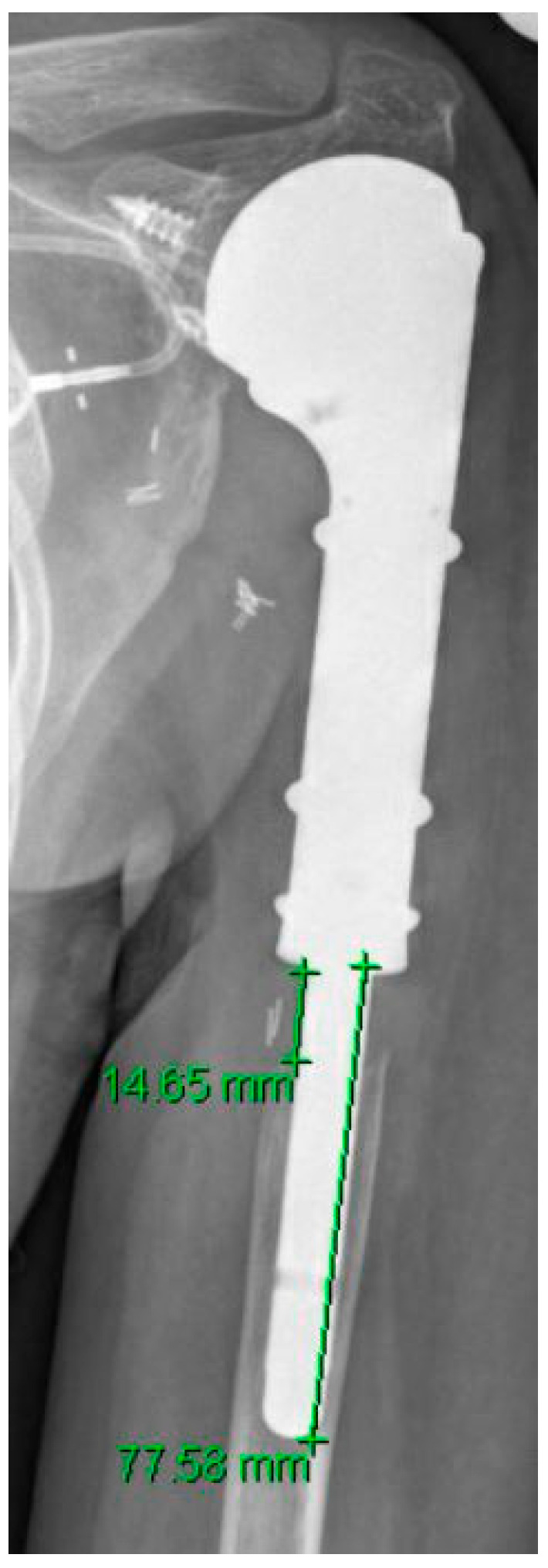
Plain radiograph of an anatomical proximal humeral replacement (hemiarthroplasty) after resection of primary sarcoma. The extent of longitudinal bone resorption (stress shielding) is measured and related to the length of the stem (14.65/77.56 × 100 = 19%). On the glenoid side, a screw anchor is visible for fixation of the attachment tube. In state after appropriate surgical measures vascular clip come to display.

**Table 1 jcm-10-03424-t001:** Patient demographics and study specifics.

Variable	*n* (%)
Included patients	65
Median follow-up	36m (IQR 18–80)
Deaths during observation	24 (32%)
Female	25 (38%)
Male	40 (62%)
Age	18 y (IQR 14–42)
Tumor histology	
Osteosarcoma	28 (43%)
Chondrosarcoma	16 (25%)
Ewing’s sarcoma	11 (17%)
Malignant fibrous histiocytoma	5 (8%)
Others	5 (8%)
Extraarticular resection	25 (39%)
Intraarticular resection	25 (38%)
Size of humeral stem	75mm
Reverse humeral replacement	15 (23%)
Hemiarthroplasty	50 (77%)
Chemotherapy	43 (66%)
Radiotherapy	16 (25%)

**Table 2 jcm-10-03424-t002:** Overview of the extent of stress shielding in different patient groups.

	Extent Stress Shielding Median (IQR)	*p* Value
All patients	14% (8–25)	
Male	14% (10–45)	0.06
Female	11% (5–21)	0.06
History of chemotherapy	16% (10–40)	0.012
No history of chemotherapy	10% (6–14)	0.012
History of radiotherapy	7% (5–21)	0.08
No history of radiotherapy	14% (9–28)	0.08
Intraarticular resection	12% (7–26)	0.43
Extraarticular resection	17% (8–24)	0.43
Reverse endoprosthesis	14% (10–29)	0.47
Anatomical hemiarthroplasty	17% (8–24)	0.47

## Data Availability

Data supporting reported results can be questioned from the author.

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
