# Peer review of "Periprosthetic Stress Shielding of the Humerus after Reconstruction with Modular Shoulder Megaprostheses in Patients with Sarcoma"

_jcm, 2021, doi:10.3390/jcm10153424_

Round 1

Reviewer 1 Report

Dear Authors, 

This is a very interesting original article. The study is well designed according to STROBE guidelines, the methods and results are well presented. The discussion is interesting and well written.

Nevertheless, I have some suggestions to improve the paper:

  1. I suggest to consider changing the expression ‘’stress shielding’’ into ‘’complete osteolysis around the stem’’. I think that using the term of stress shielding in this article can be controversial. I understand this phenomenon is seen as osteolysis, but usually, it is not complete. It is important to underline in the introduction that you are measuring only the complete osteolysis around the stem according to your original method.
  2. The line 105-107 should be corrected (the headline Patents should be removed)

Good luck with the long-term follow-up study!

Author Response

We thank the reviewers very much for their valuable comments.

The manuscript has benefited greatly from the implementation of the comments and recommendations.

We have revised our manuscript along the commentaries.

Reviewer 2 Report

General comments

The article analyses the evolution of megaprostheses and specifically the radiologic stress shielding and is potential causes (treatment, patients) and consequences (follow up, revision..).

The authors must be congratulated for their important work and those data must be shared.

Few improvements must be done to make the article more readable.

Several results observations must be rewritten (for instance chemotherapy or age impacts) as they cannot be assumed but the authors can discuss them in the discussion.

Specific comments

Table I must be more organized with subtitles (Diagnosis, treatment…) and more exhaustive

Age

Follow up

Deaths

Size of the stem (even if written in the text)

Figure 2: authors must specify what the screw in the glenoid side correspond to.

Results section should be better organized:

1/ It shoulder begin by a section “complications and revisions” with a Kaplan Meyer analysis of survival.

2/ A second section must present the clinical results of the non revised patients (ASES, pain, mobility..)

3/ A third section must be related to the topic of the article: the stress shielding and osteolysis.

4/ the influence of each parameter (age, chemotherapy..) must be studied in a fourth section. I recommend adding a table with the OddRatio of each studied parameter.

Line 180: p=0.06 is not significant (as defined in the statistical section), thus, authors cannot assume the influence of age and radiotherapy.

Discussion:

Line 212: not really true though

Line 319: same. Cannot conclude that.

Author Response

(The authors gave the same response as above.)
